# Lung Spatial Profiling Reveals a T Cell Signature in COPD Patients with Fatal SARS-CoV-2 Infection

**DOI:** 10.3390/cells11121864

**Published:** 2022-06-07

**Authors:** Chen Xi Yang, Michael Tomchaney, Manuel F. Landecho, Borja R. Zamacona, Marta Marin Oto, Javier Zulueta, Joshua Malo, Steve Knoper, Marco Contoli, Alberto Papi, Dragoş M. Vasilescu, Maor Sauler, Christof Straub, Cheryl Tan, Fernando D. Martinez, Deepta Bhattacharya, Ivan O. Rosas, Farrah Kheradmand, Tillie-Louise Hackett, Francesca Polverino

**Affiliations:** 1Centre for Heart and Lung Innovation, St. Paul’s Hospital, Vancouver, BC V6T 1Z4, Canada; yolanda.yang@hli.ubc.ca (C.X.Y.); dragos.vasilescu@hli.ubc.ca (D.M.V.); tillie.hackett@hli.ubc.ca (T.-L.H.); 2Department of Anesthesiology, Pharmacology and Therapeutics, University of British Columbia, Vancouver, BC V6T 1Z4, Canada; 3Asthma and Airway Disease Research Center, College of Medicine, University of Arizona College of Medicine, Tucson, AZ 85719, USA; michael.tomchaney@gmail.com (M.T.); fdmartin@arizona.edu (F.D.M.); 4Department of Internal Medicine, Clinica Universidad de Navarra, 31008 Pamplona, Spain; mflandecho@unav.es (M.F.L.); brecalde@unav.es (B.R.Z.); mmarin@copdnet.org (M.M.O.); 5Mount-Sinai Hospital, New York, NY 10029, USA; javier.zulueta@mountsinai.org; 6Department of Surgery, University of Arizona, Tucson, AZ 85719, USA; jmalo@arizona.edu (J.M.); sknoper@email.arizona.edu (S.K.); 7Pulmonary Division, University of Ferrara, 44121 Ferrara, Italy; ctm@unife.it (M.C.); ppa@unife.it (A.P.); 8Department of Pathology and Laboratory Medicine, University of British Columbia, Vancouver, BC V6T 1Z4, Canada; 9Department of Medicine, Yale University, New Haven, CT 06520, USA; maor.sauler@yale.edu; 10Nanostring Technologies, Seattle, WA 98109, USA; cstraub@nanostring.com (C.S.); ctan@nanostring.com (C.T.); 11Department of Immunobiology, University of Arizona, Tucson, AZ 85719, USA; deeptab@arizona.edu; 12Department of Medicine and Center for Translational Research, Baylor College of Medicine, Houston, TX 77030, USA; ivan.rosas@bcm.edu (I.O.R.); farrahk@bcm.edu (F.K.)

**Keywords:** COVID-19, chronic obstructive pulmonary disease (COPD), T cells

## Abstract

People with pre-existing lung diseases such as chronic obstructive pulmonary disease (COPD) are more likely to get very sick from SARS-CoV-2 disease 2019 (COVID-19). Still, an interrogation of the immune response to COVID-19 infection, spatially throughout the lung structure, is lacking in patients with COPD. For this study, we characterized the immune microenvironment of the lung parenchyma, airways, and vessels of never- and ever-smokers with or without COPD, all of whom died of COVID-19, using spatial transcriptomic and proteomic profiling. The parenchyma, airways, and vessels of COPD patients, compared to control lungs had (1) significant enrichment for lung-resident CD45RO^+^ memory CD4^+^ T cells; (2) downregulation of genes associated with T cell antigen priming and memory T cell differentiation; and (3) higher expression of proteins associated with SARS-CoV-2 entry and primary receptor ubiquitously across the ROIs and in particular the lung parenchyma, despite similar SARS-CoV-2 structural gene expression levels. In conclusion, the lung parenchyma, airways, and vessels of COPD patients have increased T-lymphocytes with a blunted memory CD4 T cell response and a more invasive SARS-CoV-2 infection pattern and may underlie the higher death toll observed with COVID-19.

## 1. Introduction

Severe acute respiratory syndrome coronavirus 2 (SARS-CoV-2) is the cause of the coronavirus disease 2019 pandemic, which has infected over 175 million people and caused over six million deaths to date [1]. The fatality rate for coronavirus disease 2019 (COVID-19) is estimated at 0.5–1%, with most people dying from respiratory failure related to diffuse acute lung injury and acute respiratory distress syndrome (ARDS) associated with ineffective viral clearance, especially in elderly individuals [2]. There is much debate whether smoking or other comorbidities such as chronic obstructive pulmonary disease (COPD) increase susceptibility to ARDS from COVID-19. Several studies have shown a wide range in the prevalence (1.1–38%) of COPD patients becoming infected with COVID-19 [3,4]. COPD patients are more likely to be hospitalized, admitted to the intensive care unit (ICU), and receive mechanical ventilation versus non-COPD COVID-19 patients [4]. Additionally, COPD is associated with increased susceptibility to respiratory virus infections, which are one of the most common causes of acute exacerbations of COPD [5]. In smokers with COPD, there is well-documented bystander activation of the T lymphocytes [6,7,8]. Interestingly, severe COVID-19 disease has been associated with low-avidity T lymphocyte responses to SARS-CoV-2 [9]. However, there is great uncertainty about whether the timing, composition, or magnitude of the innate and adaptive immune responses to SARS-CoV-2 are protective or pathogenic [10,11] and on how cigarette smoke and COPD shape the immune responses during SARS-CoV-2 infection. An understanding of the adaptive immune responses to SARS-CoV-2 in smokers with and without COPD is, therefore, essential to establish treatment strategies.

The goal of this study was to spatially profile the transcriptome and proteome of innate and adaptive immune cell responses within the lung parenchyma, airways and vessels of COPD patients compared to never- and ever-smokers who died of COVID-19, to develop a biological landscape of the lung immune response from a structural, immunological, and clinical standpoint.

## 2. Materials and Methods

Additional information is in the Online Appendix A.

### 2.1. Study Population

The study included seven never-smokers and eleven ever-smokers with and without COPD who died from COVID-19 pneumonia following SARS-CoV-2 infection at the University of Navarra Hospital, Pamplona, Spain (Table 1). All patients ceased smoking ≥1 year prior to hospitalization. The study conformed to the Declaration of Helsinki and was approved by the University of Arizona ethics committee (IRB #1811124026). COPD patients had a prior physician diagnosis. Computed tomography (CT) scans were analyzed for tissue volume and emphysema (% lower attenuation areas [LAA] <−950 Hounsfield units [HU]), using Vida Vision software (Version 2.20, Vida Diagnostics, Coralville, IA, USA). For each subject, the nucleocapsid (N) and envelope (E) SARS-CoV-2 gene expression levels were quantified in nasal swabs and blood using rt-PCR.

In order to assess whether the up/downregulation of proteins in COPD patients with COVID-19 was associated with SARS-CoV-2 infection or with the presence of underlying COPD, a second cohort was studied of uninfected never-smokers (*n* = 7) and ever-smokers with or without COPD (*n* =11) [12], who underwent thoracic surgery at the University of Arizona [6] before the SARS-CoV-2 pandemic started (Appendix A).

### 2.2. Spatial Gene and Protein Profiling

In patients who died of COVID-19, lung biopsies were obtained through a 5 cm incision between the 3rd and 5th intercostal space between the midclavicular and midaxillary lines at the time of death, and samples were formalin-fixed and paraffin-embedded (FFPE). Lung sections were prepared for use with the GeoMX digital spatial profiler (DSP) and the GeoMX Transcriptome and COVID-19 Immune Response panels, according to the manufacturer’s instructions (NanoString, WA, USA). For each lung section, we randomly selected 16 total regions of interest (ROIs) from the parenchyma, airways, and vessels, uniformly spread throughout the section to ensure sample randomization. Each ROI was subdivided into compartments based on fluorescent cell-specific markers, and serial UV illumination of each compartment was used to sequentially collect the probe barcodes from the different cell types as described elsewhere [13]. Once all of the indexing oligonucleotides were collected into a 96-well plate, they were counted using next-generation sequencing. For each ROI, the indexing oligonucleotides were counted using an Illumina’s i5xi7 dual-indexing system, and cDNA libraries were pair-end sequenced. Consecutive lung sections were incubated with a GeoMX DSP 41 oligo-labeled antibody panel (see Appendix A), to spatially profile proteins in the same ROIs.

### 2.3. Immunofluorescence Staining

In order to validate the main spatial proteomic findings, double immunofluorescence staining for ACE2 or CD45RO, CD4 and CD8, and epithelial cell adhesion molecule (EPCAM) or endothelial cell marker (von Willebrand factor) was performed on consecutive lung sections from the COVID-19 cohort. The number of positive cells were counted by two blinded observers and normalized using Metamorph software [14,15].

### 2.4. Statistics

The patient’s demographic data were assessed using a Kruskal–Wallis H test for continuous variables and a Fisher’s exact test for categorical variables. The days of survival from day one of COVID-19 symptoms and day of death in hospital was compared using a pair-wise Mann–Whitney U test. Differential protein and RNA expression analyses were performed to compare disease conditions (never-, ever-smokers, or COPD) or tissue structures (parenchyma, airways, or vessels), using generalized estimating equations (GEEs) to account for multiple ROIs per subject and adjusted for age using the R computing environment (Version 3.5.0, https://www.r-project.org/; accessed on 1 May 2019). Multiple hypothesis testing in all the analyses was controlled using the Benjamini–Hochberg procedure. A false discovery rate (FDR) of <0.05 was used as the significance threshold. The cellular proportions of the immune cell types between patient groups were inferred using the R package “SpatialDecon”. “SpatialDecon” is a reference-based deconvolution method that is designed specifically for the NanoString GeoMx platform. By using log-normal regression and modeling background noise, “SpatialDecon” outperformed classical least-squares methods. “SpatialDecon” provides predefined cell profile matrices generated using published single-cell RNA-seq for 75 tissue types. In our study, we performed the deconvolution analysis using the “Lung plus neutrophil” panel. Immunohistochemical staining was assessed using Kruskal–Wallis and Mann–Whitney tests, and *p* < 0.05 was considered significant.

## 3. Results

### 3.1. Study Cohort

The COVID-19 cohort demographics are reported in Table 1. There were no differences in age or biological sex between the never- and ever-smoker controls and COPD patient groups; however, due to the large variation in age of the COPD group, RNA and protein analyses were adjusted for age. COPD patients (purple) with SARS-CoV-2 infection died within a mean of 14.0 days following initial symptoms, compared to ever-smoker controls (green), who died within 17.3 days (*p* = 0.78), and never-smokers (yellow), who died within 25.6 days (*p* = 0.03) (Figure 1A). The gene expression analysis for SARS-CoV-2 Gene N (nucleocapsid) and Gene E (envelope) showed no significant differences in the viral load on hospital admission between the never- and ever-smoker controls and patients with COPD (Figure 1B,C). The high-resolution CT coronal and sagittal views of a representative never-smoker, ever-smoker, and patient with COPD, four days before death demonstrated extensive patchy consolidation, extensive ground-glass opacities, interlobular septal thickening, and interlobular pleural thickening in the lungs, consistent with changes found in ARDS (Figure 1D) [16]. The CT scans confirmed that the patients with physician-diagnosed COPD had a greater percentage of emphysema (%LAA_950_), compared to the never- and ever-smoker controls (Appendix A).

### 3.2. Spatial Transcriptomic Profiling of Immune Response Genes in COVID-19

In the COVID-19 cohort, the lung tissue sections were stained with pan-cytokeratin (epithelial marker), CD45 (leukocyte marker), and SYTO13 (DNA marker) to enable sampling of parenchyma, airways, and vessels (Figure 2A). We confirmed across the multiple ROIs sampled, that the parenchyma, airways, and vessels expressed specific gene signatures for each structure (FDR < 0.05, Appendix A). Of the total of 1839 genes, 1803 were reliably measured across the dynamic range of gene expression panels (Appendix A).

The parenchyma of COVID-19 COPD patients had greater expression of 246 genes compared to never-smokers, and greater expression of 186 genes compared to ever-smoker controls (FDR < 0.05, Figure 2B,C). Notably, these included *ACE2* and *TMPRSS2,* which cleaves the SARS-CoV-2 spike protein to allow entry into host cells [17], and *ORF1ab* encoding for the SARS-CoV-2 replicase [18]. Further, in the same comparison, we identified 71 and 92 downregulated genes compared to never-smokers and ever-smokers, respectively (FDR < 0.05), including *IL2RA* and *HLA-DR*, markers of antigen-specific activated T cells [19,20], and *IRF-7* and *IRF-9*, main regulators of type-I interferon-dependent immune responses to virus [21,22].

The airways of COVID-19 COPD patients had greater expression of 129 genes compared to never-smokers, and greater expression of 58 genes compared to ever-smoker controls (FDR < 0.05, Figure 2D,E). These included *ACE2* and *ORF1ab*. In the same comparison, we also identified the downregulated expression of 40 and 49 genes, compared to never-smokers and ever-smokers, respectively (FDR < 0.05). These included *ID2*, which allows the antigen-driven differentiation of memory T cells [23], and *SOD2*, which clears mitochondrial reactive oxygen species [24].

The pulmonary blood vessels of COVID-19 COPD patients had greater expression of 78 genes compared to never-smokers, and greater expression of 47 genes compared to ever-smoker controls (FDR < 0.05, Figure 2F,G). Notably, the same *ACE2* and *ORF1ab* genes that were increased in the airways were also upregulated in the pulmonary blood vessels. Additionally, in the same comparison, we identified the downregulated expression of 44 and 37 genes, compared to never-smokers and ever-smokers, respectively (FDR < 0.05). These included *DUSP-1*, which is thought to lead to steroid resistance in SARS-CoV-2 infection [25], and *LAIR-2*, an inhibitory immune receptor expressed by T cells [26]. There were less than 10 differentially expressed genes between never- and ever-smoker controls in the lung parenchyma, airways, and vessels (Appendix A).

We next identified the overlap of up- and downregulated genes found in the parenchyma, airways, and vessels of COPD patients compared to ever- and never-smoker patients (Figure 2H,I). The alteration in gene expression in COPD lung tissue compared to never- and ever-smoker controls was not due to increased viral load, as the SARS-CoV-2 gene *N* and *E* expression (Appendix A) was not different among the groups.

### 3.3. Protein Validation of Transcriptomic Profile in Parenchyma, Airways and Vessels

In the COVID-19 cohort, we next performed GeoMX spatial proteomic profiling for the same ROIs on a consecutive FFPE tissue section to validate the transcriptomic findings. We confirmed that ACE2, one of the most upregulated genes, was also elevated at the protein level in the airways and vessels of COPD patients compared to never- and ever-smoker controls (Figure 3A). Immunofluorescence staining of alveolar and airway epithelial cells and endothelial cells further confirmed elevated ACE2 protein expression in the parenchyma (Figure 3B) and airways (Figure 3C) of COPD patients compared to never- and ever-smoker controls. Protein expression of ACE2 was also elevated in the vessels of ever-smoker compared to never-smoker controls (Figure 3D).

### 3.4. Identification of Memory T Lymphocytes in Response to COVID-19 in COPD Patients

In the COVID-19 cohort, gene ontology pathways identified immune pathways involved in the antigen processing and presentation of endogenous peptide antigen (FDR < 0.05) enriched in the differentially expressed genes between controls and COPD patients. Thus, in order to assess the immune cell composition within the parenchyma, airways, and vessels of COVID-19 never- and ever-smoker controls and COPD patients, a lung cell deconvolution analysis was used (Figure 4A). The cell deconvolution analysis identified an increased number of T lymphocytes in the lung parenchyma, airways, and vessels of COPD patients compared to never-smoker controls (Figure 4B). The T lymphocyte proportion within the lung was negatively correlated with the absolute T lymphocyte count in the blood of patients admitted for COVID-19 (Figure 4C), suggesting increased T lymphocyte recruitment from the blood into the lung in response to SARS-CoV-2 infection, with the highest recruitment being in COPD patients.

To validate the cell deconvolution data, the GeoMX spatial proteomic profiling of T lymphocyte markers was performed in the COVID-19 cohort. Interestingly, of all the T lymphocyte protein markers measured, CD45RO was the only one upregulated in the COPD lung tissue structures compared to never- and ever-smoker controls (Appendix A). Most naïve human T lymphocytes express CD45RA, whereas CD45RO is a CD45 isoform specifically expressed on memory T lymphocytes. Spatial profiling of the CD45RO protein revealed that COPD patients had increased expression of CD45RO in the parenchyma, airways, and vessels compared to both ever- and never-smoker controls (Figure 5A). Similarly, CD45RO protein expression was also elevated in the parenchyma, airways, and vessels of ever-smokers compared to never-smokers, indicating a possible effect of smoking on the lung tissue that persists in former smokers (Figure 5A). Further, we found a positive correlation between CD45RO and ACE2 protein in the parenchyma (rho = 0.359, *p* < 0.01), airways (rho = 0.549, *p* < 0.001), and vessels, (rho = 0.474, *p* < 0.001) (Appendix A). Interestingly, the great majority of the CD45RO^+^ T lymphocytes were CD4^+^, whereas a minority were CD8^+^ (Appendix A).

We next validated our findings using immunofluorescence staining of alveolar and airway epithelial cells and endothelial cells, which confirmed elevated CD45RO protein expression in the parenchyma (Figure 5B), airways (Figure 5C), and vessels (Figure 5D) of COVID-19 COPD patients and ever-smokers, compared to never-smokers. Immunofluorescence staining of consecutive lung sections confirmed that the majority of CD45RO^+^ cells were CD4^+^ T lymphocytes (not shown). 

To determine whether the changes observed in COPD patients who died of COVID-19 were caused by SARS-CoV-2 infection or to the underlying inflammation in COPD, the same protein panels were assessed in a second cohort of lung tissue collected prior to the pandemic from never- and ever-smokers with and without COPD (see Appendix A for demographics). Surprisingly, CD45RO expression was downregulated in the parenchyma, airways, and vessels of non-COVID-19 COPD patients compared to never-smokers and the airways of ever-smoker controls (Appendix A). These data suggest that the presence of both SARS-CoV-2 infection and pre-existing COPD underlie the increase in memory T cells observed in the lungs of COPD patients who died of COVID-19.

## 4. Discussion

We report the first spatial transcriptomic and proteomic profiling in the lung parenchyma, airways, and vessels of fatal COVID-19 patients with COPD versus never-smoker and ever-smoker controls. These data indicate that COPD patients have lower survival despite similar levels of lung and blood SARS-CoV-2 viral load, which is associated with an upregulated inflammatory response involving increased numbers of memory T lymphocytes, mainly CD4^+^, but downregulation of genes associated with antigen-dependent T lymphocyte differentiation. These findings have important implications for the treatment of COPD patients with SARS-CoV-2 infection.

The T lymphocytes, CD4^+^ and CD8^+^, are key players in the adaptive immune responses triggered by cigarette smoke exposure and in COPD onset and progression, and their numbers are increased in peripheral lung tissues from COPD patients [27,28]. Given the essential role of T lymphocytes during viral infections, it is crucial to understand the nature of the immune responses to SARS-CoV-2 within the COPD lung and how the presence of pre-existing COPD can shape the immune responses to the virus. Severe COVID-19 disease is associated with a reduced proportion of activated T lymphocytes in peripheral blood and increased numbers of T lymphocytes in the lung [29]. Increased numbers of CD4^+^ and CD8^+^ T lymphocytes have been detected in patients who recovered from SARS-CoV-2 infection after having diffuse lung disease [10]. In particular, lung-resident memory T lymphocytes positioned within the respiratory tract are required to limit SARS-CoV-2 spread and COVID-19 severity [30]. Tissue-resident memory T lymphocytes’ responses affect the susceptibility to and the pathogenesis of SARS-CoV-2 infection by secreting pro-inflammatory interferon (IFN) γ and anti-inflammatory interleukin (IL)-10 molecules [31]. We observed an increased memory T, in particular CD4^+^, lymphocyte immune infiltration within the lungs of COPD patients, compared with ever- and never-smoker controls, all of whom died of COVID-19, which was associated with an upregulation of crucial genes associated with SARS-CoV-2 entry and infection (e.g., *ACE2*, *TMPRSS*, and *ORF1ab*). Interestingly, the increase in the memory T lymphocyte population, confirmed both by spatial proteomics data and immunohistochemical staining, was accompanied by the downregulated expression of genes associated with memory function and antigen-driven maturation. To confirm that our findings were associated with SARS-CoV-2 infection and not the pre-existing immune milieu associated with COPD itself, CD45RO staining was performed on a separate cohort of never-smokers, ex-smokers, and patients with COPD that underwent lung resection surgery prior to the COVID-19 pandemic. In this tissue cohort, in the absence of SARS-CoV-2 infection, the numbers of memory T lymphocytes in COPD parenchyma, airways, and vessels were reduced, compared to ever- and never-smoker controls. These data together indicate that the increase in memory T lymphocytes observed in the COVID-19 COPD patients is specific to SARS-CoV-2 infection.

Lung-resident memory T lymphocytes in host immunity are crucial for mucosal responses against pathogens including SARS-CoV-2 (e.g., in airways and interstitial tissue), and limit re-infections locally [30]. In stable COPD, the T lymphocyte responses may be impaired [7,32,33], thus explaining the lower levels of memory T lymphocytes in uninfected COPD patients. However, during a viral challenge such as SARS-CoV-2 infection, the memory T lymphocytes, although upregulated as suggested by elevated CD45RO, may be off-targeted in COPD, thus explaining the downregulation of key genes involved in memory T lymphocyte functioning and antigen priming observed in COVID-19 COPD patients. Our findings are in line with recent reports indicating that SARS-CoV-2 and other beta coronaviruses induce multi-specific and long-lasting T lymphocyte immunity responses with strong upregulation of memory T lymphocytes against the viral structural proteins [31]. However, it is important to note that the T lymphocyte immune profile in COPD lungs was different from the one in never- and ever-smoker controls. Specifically, we found increased numbers of memory T lymphocytes but reduced expression of key genes involved in antigen-specific T lymphocyte priming and regulation of type-I interferon-dependent immune responses to the virus, suggesting off-targeted lung-resident memory T lymphocyte responses to SARS-CoV-2 in COPD. Our finding that the great majority of CD45RO^+^ lymphocytes were CD4^+^ is in line with previous studies showing that memory CD4^+^ responses to SARS-CoV-2 infection are somewhat larger than those of the CD8^+^ pool, potentially reflecting antigen persistence [34]. SARS-CoV-2 infection and disease are regulated by a fine balance between CD4 and CD8 T lymphocyte subsets. If, on one hand, high expression levels of effector and memory CD8^+^ T cells in acute COVID-19 are associated with effective SARS-CoV-2 clearance and improved clinical outcomes, higher expression of memory CD4 compared to CD8 T lymphocytes has been associated with impaired T cell responses to SARS-CoV-2 [34], which might contribute to the poor clinical outcome observed in COVID-19 COPD patients.

Upregulation of the memory T lymphocytes was associated with upregulation of ACE2 protein levels measured by spatial protein profiling and immunostaining. Although the upregulation of ACE2 has been reported in COPD lungs by several studies, to the best of our knowledge, this is the first study that investigated the levels of ACE2 in the lung parenchyma, airways, and vessels of ever- and ex-smokers and patients with COPD who died from COVID-19. Interestingly, we found a positive association between ACE2 and CD45RO protein levels across all the three lung structures studied. These results suggest that the ACE2-dense lung regions, that likely represent the SARS-CoV-2 entryways into the lung, may represent immune “hot spots” where T lymphocytes are recruited from the bloodstream (where we showed reduced cell numbers), into the lung where they are challenged by the viral antigens, and develop a memory phenotype.

Previous studies in individuals who died of COVID-19 infection showed progressively reduced expression of ACE2 in the lower airways, with limited expression in the alveoli, paralleled by a striking gradient of SARS-CoV-2 infection in the proximal (high) versus distal (low) pulmonary epithelium [35]. However, the spatially resolved data in this report shows for the first time that the alveolar epithelium of patients with COPD is the region where COPD patients who died of COVID-19 had the highest number of differentially expressed genes compared to never- and ever-smoker controls, including *ACE2* and *TMRPSS2*, while showing blunted memory T lymphocyte responses. This important piece of information indicates that, although the upper airways might be the main sites of SARS-CoV-2 entry and infection [35], viral infection triggers a cascade of events that spread into the distal lung structures, in particular the alveolar epithelium, which is reduced in emphysematous lungs, and is associated with a detrimental immune activation and may cause higher mortality in COPD patients. It is important to note that no difference between the groups was found in the use of systemic corticosteroids or any anti-COVID therapies received during the hospitalization period, indicating that the changes in the T lymphocytes found in COPD patients are unlikely to be caused by different underlying therapies.

This study has limitations to note: first, the sample size of the COVID-19 patients is small, due to ongoing challenges that limit the access to tissue from these patients. However, spatial profiling generates a high amount of multiplexed information on different regions of interest within the lung structure. Second, recent spirometry data were lacking for 15 out of 18 subjects as most of the COVID-19 patients were admitted in critical condition. However, the patients had a prior clinical diagnosis of COPD, and emphysema was confirmed on the CT scans of the COPD patients. Third, previous cigarette smoke exposure can affect the gene expression profile within the lung [36]. However, all our study subjects ceased smoking at least one year prior to the study, and there was no difference in pack/years between COPD and ever-smoker controls, which minimizes the effect of cigarette smoke on the transcriptomic data. Fourth, it is difficult to distinguish how much of the immune profile found in COPD patients versus controls was driven by the presence of COPD and how much it was driven by the presence of SARS-CoV-2 infection. To address this, we profiled CD45RO expression in a non-infected cohort of never- and ever-smokers with and without COPD, these data confirmed that the upregulation of CD45RO expressing memory T lymphocytes in COPD patients who died of COVID-19 was likely driven by SARS-CoV-2 infection. Last, our main study cohort is skewed toward the most severe cases of SARS-CoV-2 infection, i.e., the ones who died of COVID-19. This was due to the fact that lung autopsy is the only source of COVID-19 lung tissue for research, and it is not possible to study lung biopsies from patients with milder COVID-19 symptoms who survived.

## 5. Conclusions

We report for the first time that, within the lung parenchyma, airways, and vessels of COPD patients with SARS-CoV-2 infection, there is a significant enrichment for memory CD4 T lymphocytes with a blunted memory phenotype, associated with an increase in genes crucial for SARS-CoV-2 entry and infection. Using spatial transcriptomic and proteomic characterization of parenchyma, airways, and vessels provides a biologically interpretable landscape of COVID-19 lung pathology and serves as an important resource for future treatment of patients with COPD and SARS-CoV-2 infection.

## Figures and Tables

**Figure 1 cells-11-01864-f001:**
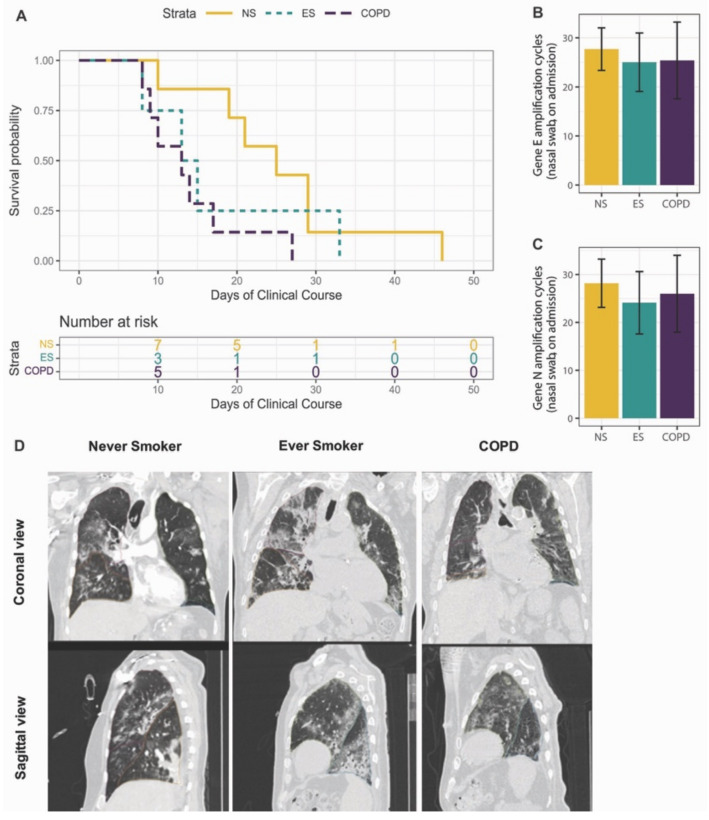
COVID-19 patient characteristics: (**A**) Survival curve for the days of the clinical course of never-smokers (NS), ever-smokers (ES), and COPD patients infected with severe acute respiratory syndrome coronavirus 2 (SARS-CoV-2). The survival analysis was performed using day one of COVID-19 symptoms and day of death in hospital, using a log-rank test. (**B,C**) The gene expression levels of the envelope (E) (**B**) and nucleocapsid (N) (**C**) SARS-CoV-2 genes were quantified in nasal swabs. (**D**) Thoracic computed tomography (CT) coronal and sagittal views in a representative never smoker (left panel), ever-smoker (middle panel), and patient with COPD (right panel) within four days before death due to SARS-CoV-2 infection.

**Figure 2 cells-11-01864-f002:**
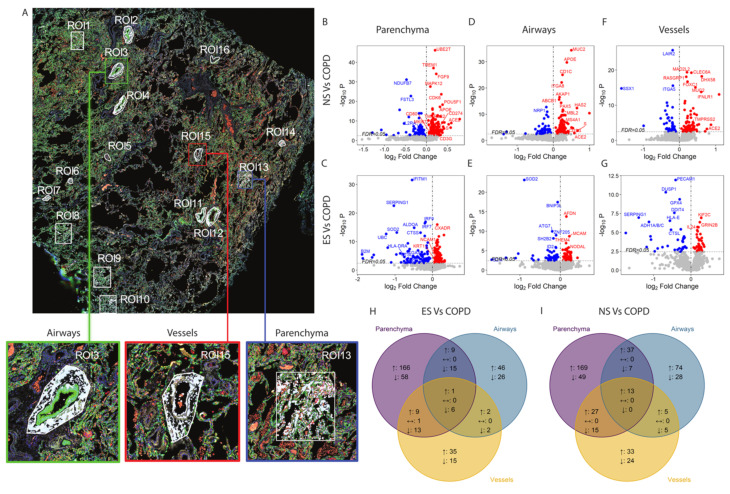
NanoString GeoMX spatial transcriptomic profiling: (**A**) Image of a lung tissue section from a COPD subject stained with pan-cytokeratin (green) to identify epithelial cells, CD45 (red) to identify lymphocytes, and SYTO13 (blue) to identify nuclei. Sixteen randomly selected regions of interest (ROIs) were sampled for spatial protein and RNA analysis and are highlighted in white. The higher-magnification inserts show representative airways (green), vessels (red), and parenchyma (blue) structures selected as ROIs. (**B**–**G**) Volcano plots showing the genes expressed in the lung parenchyma (**B**,**C**), airways (**D**,**E**), and vessels (**F**,**G**) of patients with COPD who died of COVID-19 versus never-smoker controls (NS, panels **B**,**D**,**F**) and ever-smoker controls (ES, panels **C**,**E**,**G**). The data were assessed using generalized estimating equations (GEEs) accounting for multiple ROIs per case and were adjusted for age. The red dots indicate upregulated genes and the blue dots downregulated genes in COPD patients versus never- and ever-smokers for the specific tissue structures. The horizontal dotted line indicates the significance threshold of FDR < 0.05, and the vertical dashed line indicates the effect size of 0. The gray dots indicate genes that were not significant at FDR < 0.05. The top genes ranked by *p*-value are labeled with their gene symbols. (**H**,**I**) Venn diagram comparing the transcriptomic signatures in the parenchyma, airways, and vessels structures in COPD vs. ever-smoker controls (**H**) and vs. never-smoker controls (**I**). The up/down arrows indicate genes that are up/downregulated in the specific structure, and the double-sided arrows indicate markers with the inconsistent direction of change between tissue structures.

**Figure 3 cells-11-01864-f003:**
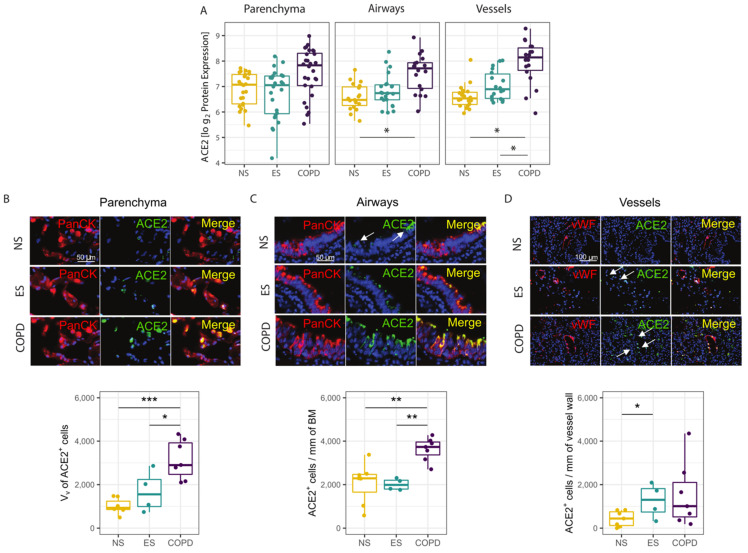
ACE2 spatial proteomic expression: (**A**) ACE2 protein expression levels were quantified in the parenchyma, airway, and vessel ROIs in formalin-fixed paraffin embedded (FFPE) lung tissue sections from patients with COPD versus ever- (ES) and never-smoker (NS) controls by NanoString GeoMX digital spatial profiling. The data were assessed using generalized estimating equations (GEEs) accounting for multiple ROIs per case and were adjusted for age. The box plots represent the median and interquartile range, the error bars are the 5th and 95th percentile, and each dot represents a single ROI. (**B**–**D**) ACE2^+^ cells were quantified in the parenchyma (**B**), airways (**C**), and vessels (**D**) in patients with COPD versus ever- (ES) and never-smoker (NS) controls by double immunofluorescence staining for ACE2 (green) and epithelial (pan-cytokeratin, red) or endothelial cells (von Willebrand factor, red). The non-normally distributed data were assessed using Kruskal–Wallis and Mann–Whitney tests, the box plots represent the median and interquartile range, the error bars indicate the 5th and 95th percentile, and each dot represents a single subject. * Indicates *p* < 0.05, ** indicates *p* < 0.01, ***: *p* < 0.001.

**Figure 4 cells-11-01864-f004:**
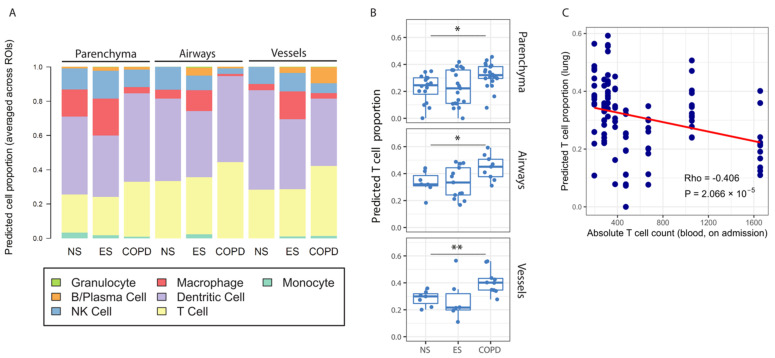
Deconvolution of spatial transcriptomics and T cell levels in the blood. (**A**) The cellular proportions of the immune cell types between tissue structures (parenchyma, airways, and vessels) and disease conditions (NS, ES, and COPD) were inferred using the R package “SpatialDecon”, a deconvolution method developed for the NanoString GeoMX RNA assay. (**B**) The predicted number of T cells between tissue structures (parenchyma, airways, and vessels) in never- and ever-smokers with and without COPD. (**C**) Inverse correlation between predicted cell proportion in the lung obtained by cell deconvolution and absolute T cell count in blood on admission. * Indicates *p* < 0.05, ** indicates *p* < 0.01.

**Figure 5 cells-11-01864-f005:**
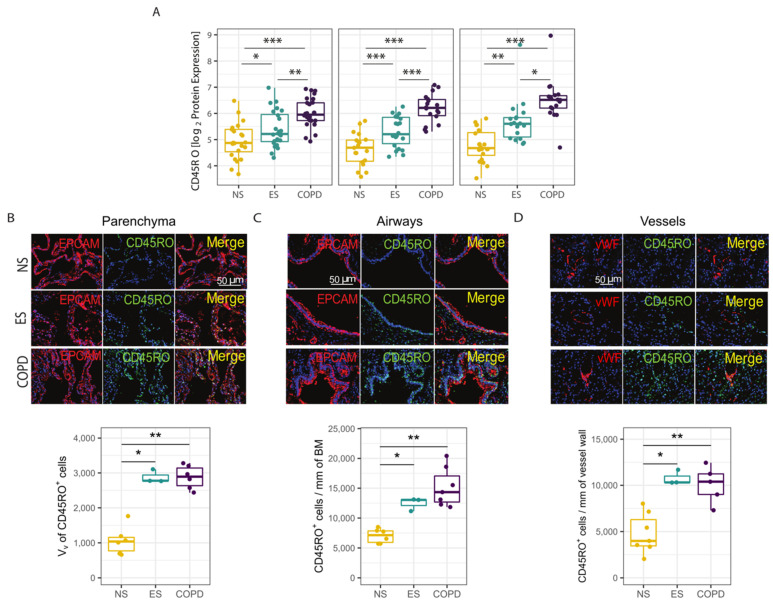
CD45RO spatial protein expression. (**A**) CD45RO protein expression levels were quantified in the parenchyma, airway, and vessel ROIs in formalin-fixed paraffin-embedded (FFPE) lung tissue sections from patients with COPD versus ever- (ES) and never-smoker (NS) controls by NanoString GeoMX digital spatial profiling. The data were assessed using generalized estimating equations (GEEs) accounting for multiple ROIs per case and were adjusted for age. The box plots represent the median and interquartile range, the error bars indicate the 5th and 95th percentile, and each dot represents a single ROI. (**B**–**D**) CD45RO^+^ cells were quantified in the parenchyma (**B**), airways (**C**), and vessels (**D**) in patients with COPD vs. ever- (ES) and never-smoker (NS) controls by double immunofluorescence staining for CD45RO (green) and epithelial (pan-cytokeratin, red) or endothelial cells (von Willebrand factor, red). The non-normally distributed data were assessed using Kruskal–Wallis and Mann–Whitney tests, the box plots represent the median and interquartile range, the error bars indicate the 5th and 95th percentile, and each dot represents a single subject. *: *p* < 0.05; **: *p* < 0.01; ***: *p* < 0.001.

**Table 1 cells-11-01864-t001:** Demographic and clinical characteristics of patients who died of COVID-19.

	S	ES	COPD	*p*
Total Participants (*n*)	7	4	7	
Age	76.00 [62,83]	73.00 [69,76]	80.00 [77,88]	0.222
Gender (female)	5 (71.4)	3 (75.0)	1 (14.3)	0.085
Smoking history (former)	0 (0)	4 (100.0)	7 (100.0)	0.001
**Comorbidities**				
Hypertension	5 (71.4)	3 (75.0)	7 (100.0)	0.4
Other cardiovascular diseases	3 (42.9)	3 (75.0)	4 (57.1)	0.832
Diabetes mellitus	1 (14.3)	1 (25.0)	4 (57.1)	0.316
Chronic kidney disease	1 (14.3)	2 (50.0)	2 (28.6)	0.565
**Medications**				
ICS	0 (0.0)	0 (0.0)	2 (28.6)	0.314
LABA/SABA/LAMA	0	0	7 (100%)	<0.001
Oral corticosteroids	0 (0.0)	0 (0.0)	1 (14.3)	1
ACEi/ARB	3 (43%)	0	3 (43%)	0.123
**COVID-19 Treatments**				
Hydroxyloroquine	7 (100.0)	4 (100.0)	7 (100.0)	NA
Tocilizumab	4 (57.1)	0 (0.0)	1 (14.3)	0.188
Betaferon	0 (0.0)	0 (0.0)	1 (14.3)	1
Corticosteroids	6 (85.7)	3 (75.0)	6 (85.7)	1
Heparin	7 (100.0)	4 (100.0)	7 (100.0)	NA
Statins	6 (85.7)	2 (50.0)	6 (85.7)	0.472
d-OTI	0.00 [0.00, 0.00]	0.00 [0.00, 1.50]	8.00 [4.50, 12.50]	0.031
**Blood Gas Data**				
SpO_2_	88.00 [87.00, 90.50]	89.50 [87.75, 91.75]	93.00 [88.50, 95.00]	0.539
paO_2_	56.30 [53.90, 67.50]	60.25 [56.57, 63.92]	66.60 [64.10, 67.50]	0.820
paCO_2_	38.30 [34.25, 39.45]	31.10 [29.20, 33.00]	37.10 [32.20, 38.60]	0.406

Data are expressed as the median [interquartile range] for continuous variables and *n* (%) for categorical variables. NS: never-smokers; ES: ever-smokers without COPD; ICS: inhaled corticosteroid, LABA: long-acting beta-agonists; SABA: short-acting beta-agonists; LAMA: long-acting muscarinic agents; ACEi: ACE-inhibitors; ARB: angiotensin receptor blockers; d-OTI: days spent with orotracheal intubation; SpO_2_: blood oxygen saturation; paO_2_: partial pressure of oxygen; paCO_2_: partial pressure of carbon dioxide. The *p*-values are from the Kruskal–Wallis multi-group comparisons. *p* < 0.05 means that at least one group median is different from the rest of the groups.

## Data Availability

Data are contained within the article or Appendix A. The data presented in this study are available upon request to Francesca Polverino and Tillie-Louise Hackett.

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
