# Peer review of "Lung Spatial Profiling Reveals a T Cell Signature in COPD Patients with Fatal SARS-CoV-2 Infection"

_cells, 2022, doi:10.3390/cells11121864_

Round 1

Reviewer 1 Report

Authors wrote a very interesting study about comics profiling of the immune microenvironment of lung in never- and ever-smokers with or without COPD, whom all died for the ARDS-induced COVID-19.

Their findings led to support the hypothesis that the lung parenchyma, airways, and vessels of COPD patients have increased T-lymphocytes with a blunted memory CD4 T cell response which drove to and a more invasive SARS-CoV-2 infection pattern with higher mortality.

The strengths of the study is the careful selection of patients and the use of standardized methods

The limitations of the study are due to the relatively small sample size. 

I may suggest to add more info on covid disease, including: PCR or antigen swab/ blood gas data/if put on O2, NIV, intubation etc./mortality rate and days after negativization.

Moreover I found some English errors throughout the manuscript. Please review it one more time.

Author Response

Reviewer 1

R1C1: Authors wrote a very interesting study about comics profiling of the immune microenvironment of lung in never- and ever-smokers with or without COPD, whom all died for the ARDS-induced COVID-19.Their findings led to support the hypothesis that the lung parenchyma, airways, and vessels of COPD patients have increased T-lymphocytes with a blunted memory CD4 T cell response which drove to and a more invasive SARS-CoV-2 infection pattern with higher mortality.The strengths of the study is the careful selection of patients and the use of standardized methods

R1R1: We thank the reviewer for the positive comments on our manuscript

R1C2: The limitations of the study are due to the relatively small sample size. 

R1R2: As discussed in the manuscript, the study includes lung tissue from individuals who died of COVID-19, which are very difficult samples to obtain as most autoptic procedures have been limited due to infection risk concerns. Also, to the best of our knowledge, this is the only study assessing the protein and RNA expression profile in patients with COPD who died of COVID-19. Thus, even with a low number of subjects, we believe our data are extremely timely and important.

R1C3:  May suggest to add more info on covid disease, including: PCR or antigen swab/ blood gas data/if put on O2, NIV, intubation etc./mortality rate and days after negativization.

R1R3: We thank the reviewer for this suggestion. Firstly, we confirm that in Figures 1B-C and Supplemental Figures S3A and B of the submitted manuscript, the data of the viral load of SARS-CoV-2 gene N (nucleocapsid) and E (envelope) expression upon hospital admission and post death for each patient were already included. We have now added, in the patient demographics table 1, the days each patient was intubated (if intubation was performed), and also their SpO2, pO2and pO2 levels.

R1C4: Moreover I found some English errors throughout the manuscript. Please review it one more time.

R1R4: We have reviewed and made typographical changes where necessary.

Reviewer 2 Report

The authors demonstrate that COPD patients infected by SARS-CoV-2 have a worse prognosis because in the lung parenchyma, airways and vessels of COPD patients there is an increase in T lymphocytes with an attenuated memory CD4 T cell response and a SARS- CoV-2 infection pattern. Although it is very interesting here are some comments:

- Rewrite the abstract without the sections.

- In the keywords section remove "keyword 1, 2 and 3"

- Table 1 is badly placed, I don't know if it's because of passing it to pdf.

- In table 1 put the FEV1/FVC the same as in the non-covid. The p according to which group is it?

- They put figure 2 first than figure 1, it's confusing.

- Line 182 can be a title? If so, point 3.2 is 3.3.

- Figure 1 is also incorrectly entered.

- Is it possible to improve the quality of the images? Figure 2 doesn't look very good.

- The image of line 316 could be put in supplementary material

Author Response

Reviewer 2

R2C1: Rewrite the abstract without the sections.

R2R1: The sections have been removed from the abstract.

R2C2: In the keywords section remove "keyword 1, 2 and 3"

R2R2: The text “keywords 1, 2, and 3: have been removed.

R2C3: Table 1 is badly placed, I don't know if it's because of passing it to pdf.

R2R3: According to the journal guidelines, the table must be placed in the section of the text where it is first cited. We will discuss this with the journal as we cannot change the placement of Table 1.

R2C4: In table 1 put the FEV1/FVC the same as in the non-covid. The p according to which group is it?

R2R4: As discussed in the limitations of the submitted manuscript, spirometry data are lacking for 15 out of 18 subjects as most of the COVID-19 patients were admitted in critical condition and could not undergo pulmonary function tests. The available spirometries performed in the three patients before admission confirmed the diagnosis of COPD. Also, the patients had a prior physician-diagnosed COPD, and the presence of emphysema in the COPD patients was confirmed with HRCT scans, as well as the lack of emphysema in the controls groups. The p-values in Table 1 are from the Kruskal-Wallis multi-group comparisons. P<0.05 means that at least one group median is different from the rest of the groups. We have now included this in the figure legend.

R2C5: They put figure 2 first than figure 1, it's confusing.

R2R5: We apologize for the confusion and we now list the figures correctly within the text.

R2C6: Line 182 can be a title? If so, point 3.2 is 3.3.

R2R6: We checked the original manuscript and did not find any title or sub-title or sub-heading in line 182. The sub-headings of the Results section are: Study cohort; Spatial transcriptomic profiling of immune response genes in COVID-19; Protein validation of transcriptomic profile in parenchyma, airways and vessels; Identification of memory T lymphocytes in response to COVID19 in COPD patients. We let the journal editorial office decide the numbering of the paragraphs.

R2C7: Figure 1 is also incorrectly entered.

R2R7: Figure 1 is now listed correctly before figure 2. If the reviewer sees otherwise, this might due to some errors occurring after the submission of the manuscript.

R2C8: Is it possible to improve the quality of the images? Figure 2 doesn't look very good.

R2R8: We have improved the quality of the Venn diagrams in Figure 2 and will re-send to the journal. Please note that, as per editorial policy, the figures must be embedded in the word file when submitting the manuscript, which might have caused the lack of resolution of the figures. We have now submitted each figure separately with high resolution and will ask the journal to use these figures in the published version of the manuscript.

R2C9:  The image of line 316 could be put in supplementary material

R2R9: We believe the reviewer refers to the sentence “consecutive lung sections confirmed that the majority of CD45RO+ cells were CD4+ T lymphocytes”. Please note that this staining was made to confirm the nature of the CD45RO+ cells, but no publication-quality pictures were taken at the time of the staining. As the staining was performed in 2021, the fluorescence on the slides has now decreased and does not allow us to capture publication quality pictures.

Round 2

Reviewer 1 Report

ok for acceptance now.